# Fitting Contralateral Neuroanatomical Asymmetry into the Amyloid Cascade Hypothesis [note 1]

**DOI:** 10.3390/healthcare10091643

**Published:** 2022-08-29

**Authors:** Fernando Arreola, Benjamín Salazar, Antonio Martinez

**Affiliations:** 1Programa de Ingeniería Biomédica, Universidad de Monterrey, San Pedro Garza García 66238, Mexico; 2Departamento de Ingeniería, Universidad de Monterrey, San Pedro Garza García 66238, Mexico

**Keywords:** Alzheimer’s disease, dementia, prevention, asymmetry biomarkers, CSF biomarkers, MRI

## Abstract

Alzheimer’s Disease (AD) is the most common cause of dementia. Due to the progressive nature of the neurodegeneration associated with the disease, it is of clinical interest to achieve an early diagnosis of AD. In this study, we analyzed the viability of asymmetry-related measures as potential biomarkers to facilitate the early diagnosis of AD. These measures were obtained from MAPER-segmented MP-RAGE MRI studies available at the Alzheimer’s Disease Neuroimaging Initiative (ADNI) database, and by analyzing these studies at the level of individual segmented regions. The temporal evolution of these measures was obtained and then analyzed by generating spline regression models. Data imputation was performed where missing information prevented the temporal analysis of each measure from being realized, using additional information provided by ADNI for each patient. The temporal evolution of these measures was compared to the evolution of other commonly used markers for the diagnosis of AD, such as cognitive function, concentrations of Phosphorylated-Tau, Amyloid-β, and structural MRI volumetry. The results of the regression models showed that asymmetry measures, in particular regions such as the parahippocampal gyrus, differentiated themselves temporally before most of the other evaluated biomarkers. Further studies are suggested to corroborate these results.

## 1. Introduction

Alzheimer’s disease (AD) is characterized by progressive cognitive and functional dysfunction. Due to the fact that symptoms of AD develop gradually, an intermediate phase of cognitive decline is defined; this stage is known as Mild Cognitive Impairment (MCI). The most common biomarkers used currently to guide the diagnosis of AD are structural MRI volumetry, cerebrospinal fluid (CSF) tests for Amyloid-β1-42 (Aβ) and Phosphorylated-Tau (P-Tau) concentrations, and PET Aβ imaging; these being used in conjunction with evaluations of cognitive function [1]. It has been shown that for each phase of the disease, the sensibility of each biomarker can vary, and therefore their usefulness depends on the phase present when the patients are screened [2].

The dynamic model of temporal biomarker evolution suggested by Jack et al. proposes that as Alzheimer’s disease progresses, the biomarkers evolve from a healthy value to an abnormal value sigmoidally with time; each biomarker undergoes this evolution in different stages of the disease [2]. The model identifies Aβ and Tau concentrations, respectively, as the earliest identified neurodegenerative biomarkers to show noteworthy changes, occurring while the patient still displays normal cognitive functioning; both of these can be measured by employing either CSF samples or PET imaging. The next biomarkers that show notable abnormalities are changes in brain structure measured through structural MRI, and measurements of memory dysfunction and clinical function scored using clinical tests. According to the model, while abnormal changes in brain structure and memory measurements grow noticeable during the MCI stage of Alzheimer’s disease, changes in clinical function typically manifest once the disease has reached its dementia stage.

While no cure for AD is currently known, an early diagnosis can be of value to the patient. Since the symptoms of AD impact patients’ abilities to perform their daily life activities, an early diagnosis can aid patients by giving them and their families sufficient time to plan any necessary adjustments to the patient’s lifestyle in order to maintain a good quality of life for them. An early diagnosis can also give the patient well-timed access to treatments for the disease, either aimed at improving their quality of life or enhancing their cognition [3].

Measurements of asymmetry acquired from structural MRI studies have been proven to be sensitive biomarkers during the MCI and AD phases [4]. It is known that cortical thickness becomes affected as subjects age and AD progresses, causing an increase or decrease in asymmetry levels depending on the region. For regions such as the inferior parietal lobule, it has been suggested that a progressive decrease in the degree of asymmetry can be used as a marker for a patient that progresses from MCI to AD [5]. In addition, it has been reported that as patients progress through the different stages of AD development, they show an increase in hippocampal asymmetry and obtain worse results in cognitive tests [6]. These examples support the idea that asymmetry as a biomarker could be just as viable as hemispheric or regional volumetric measurements.

The potential of contralateral neuroanatomical asymmetry as a biomarker for the early diagnosis of AD is proposed in this study. Due to the lateralized nature of AD-related brain atrophy, we suspect that biomarkers based on morphological asymmetry could be valuable to aid in an early diagnosis of AD, potentially showing earlier sensitivity than other biomarkers commonly used for this purpose. In this study, we analyzed 40 different pairs of contralateral neuroanatomical regions from 813 subjects and found that some of them show significant changes earlier than volumetric measurements also derived from MRI.

## 2. Materials and Methods

The flowchart shown in Figure 1 illustrates the methodology used in this work. First, a morphologic asymmetry analysis was applied to the acquired MAPER-segmented MRI studies. The asymmetry-related measurements calculated through this analysis were then merged with the patient information obtained from the ADNI platform, generating a single dataset with all the information to compare and analyze. For the longitudinal characterization of the biomarkers, all patients should have the same temporal point of reference; this point of reference was chosen to be the point in time in which the patient’s diagnosis evolved from MCI to AD. For patients who never went through this time reference and maintained the same diagnosis in every visit, an additional time imputation procedure was required in order to estimate this point of reference. The imputed time value is taken from the most similar patient who went through both MCI and AD stages, this way allowing us to align both types of patients under the same time scale. This imputation process was then repeated 1000 times for these patients, randomly imputing time points from the 5 most similar patients who underwent an MCI-to-AD diagnosis, obtaining 1000 different imputed time variables as a result. A two-sided Kolmogorov–Smirnov test was performed between each of these and the original imputed time variable to evaluate if the choice of imputed time points resulted in significant differences in the distribution of the time variable used to characterize the biomarkers. Finally, a spline regression model for each marker was obtained, with the resulting knots showing the most significant inflection point for each.

For this study, a comparison between commonly used markers and asymmetry-related markers was performed by characterizing their temporal evolution.

### 2.1. Data

Data used in the preparation of this article were obtained from the Alzheimer’s Disease Neuroimaging Initiative (ADNI) (adni.loni.usc.edu). The ADNI was launched in 2003 as a public–private partnership, led by Principal Investigator Michael W. Weiner, MD. The primary goal of ADNI has been to test whether serial magnetic resonance imaging (MRI), positron emission tomography (PET), other biological markers, and clinical and neuropsychological assessment can be combined to measure the progression of mild cognitive impairment (MCI) and early Alzheimer’s disease (AD). For up-to-date information, see www.adni-info.org (accessed on 1 February 2021).

Specifically, the data downloaded from ADNI consist of MRI MP-RAGE studies, segmentation masks for these images, concentrations of β-Amyloid (Aβ), total tau (t-tau), and hyperphosphorylated tau (P-tau) obtained via samples of cerebrospinal fluid (CSF), APOE gene study results, relevant demographic information, results of the Mini-Mental State Examination (MMSE), and a composite score known as the ADNI Memory Score (ADNI-MEM), which unifies results obtained from the MMSE, Rey Auditory Verbal Learning Test (RAVLT), Alzheimer’s Disease Assessment Scale-Cognitive Score (ADAS-Cog), and Logical Memory I exams [7,8].

The segmentation masks were created using the MAPER (Multi-Atlas Propagation with Enhanced Registration) methodology, which divides the brain into 83 anatomical regions by performing an anatomical segmentation of the MRI images based on a series of atlases that encompass potential anatomical variabilities to be found in the images, as well as adding tissue classification information to the process to improve the method’s robustness [9]. In this set, there are 40 pairs of contralateral regions with respect to the longitudinal fissure of the brain. 

### 2.2. Subject Inclusion Criteria

All participants and their studies were obtained from the Alzheimer’s Disease Neuroimaging Initiative (ADNI) public database. The main goal of ADNI is to test common and novel biomarkers’ ability for early detection and tracking of Alzheimer’s progression.

For participants to be eligible for this study, they must have available information regarding their diagnostic stage, age, gender, marital status, dominant hand, years of education, work background, and the APOE-4 alleles present in the patient. Their Mini-Mental State Examination score (MMSE), and their ADNI-MEM score are also mandatory. Patients that maintain a healthy diagnosis throughout their registered observations are also excluded.

The initial database started with 2268 patients and after the process of selection, a total of 1382 patients satisfied the requirements. Of these, 828 patients also had their results of CSF tests for Aβ and P-Tau concentrations available, and 813 had MRI scans performed at baseline, which were used to generate measurements of contralateral neuroanatomical asymmetry. An overview of the measurements gathered directly from ADNI is shown in Table 1.

### 2.3. Image Processing

The methodology used to measure and store contralateral asymmetry was developed using the C++ build of the Insight Toolkit (ITK) library (Version 5.0.1) [10]. For each contralateral pair of regions, the features were acquired as follows: First, the pair of regions in the MP-RAGE image were isolated and stored as two different images by using the information from the segmentation mask. Next, a reflection transformation along the longitudinal axis was applied to the image corresponding to the region located in the left hemisphere. What follows is a rigid registration of the two images: the one with the region from the right hemisphere and the one with the reflected region from the left hemisphere; this was performed using ITK’s registration framework. The transformation tuned through this process was a rigid 3D versor transformation with mean squared error (MSE) as its metric, and regular step gradient descent as its optimizer. Once the rigid registration process was completed, the Jaccard index of the resulting regions was computed. The Jaccard index provides an estimate of similarity between the shape and size of the two regions; therefore, we utilized it as a metric of contralateral asymmetry for each pair of regions.

In addition to the Jaccard index, and in order to be able to compare these features with well-known MRI-derived biomarkers, volumes and surface areas were also computed for each region. The contralateral average and absolute difference of these measurements were calculated. This yielded a total of five MRI-related features calculated for each pair of regions: mean contralateral volume (mV), absolute difference between contralateral volumes (dV), mean contralateral surface area (mSA), absolute difference between contralateral surface areas (dSA), and contralateral asymmetry as measured by the Jaccard index (J). With 40 pairs of regions per study, this results in a total of 200 different variables to analyze longitudinally. Finally, in order to remove potential size biases, dV and dSA measurements were respectively divided by the corresponding mV and mSA measurements for each region [5].

### 2.4. Time Data Processing

What followed was the integration of the image features and other relevant information acquired from ADNI into a single dataset. This was performed using R’s dplyr package (version 1.0.7) by merging all downloaded datasets by their registration ID (RID) and visit code (VISCODE2) [11]. The visit code was used as an elementary time variable, since it indicates the amount of time (in months) that has passed since the baseline visit took place. All images analyzed in this study were taken on the baseline visit.

The reference point of the time variable acquired from each dataset was the baseline visit. However, this is an arbitrary point of reference, since the diagnosis of each subject on the baseline visit varies. As such, it was necessary to redefine the reference point for all patients. To achieve this, the first visit in which a subject progresses from MCI to AD was established as the new reference point in time for that specific subject. For example, if a subject with an MCI diagnosis at baseline receives their first AD diagnosis at the 24-month visit, that visit becomes the reference point; the baseline visit is now defined as −24 (24 months before the event), and their 36-month visit is now considered as 12 (12 months after the event). 

However, subjects who did not show this progression were not able to have their point of reference in time established this way. Out of the 1382 subjects included in the dataset, a total of 990 remain stable in one of the three previously described phases for the duration of their visits (NL, MCI, and AD); we define these as static subjects. In order to generate a point of reference in time for this subset of the population, a process of time imputing based on distance metrics was used. For static MCI subjects, the data from their last visit was compared to the data of every non-static subject at each of their visits, ranking them by similarity according to their MMSE score, ADNI-MEM score, and age. This similarity was calculated using a Euclidean distance metric, normalizing each of the three previously mentioned variables based on their value range. Then, the adjusted time of the best match was applied to the static subject. That is, for a static subject whose last visit was the 48-month visit, and whose best match was with a subject at an adjusted time of −12 months, the 48-month visit now becomes the −12-month visit and the baseline visit becomes the −60 month visit. A similar process was carried out for static AD subjects, but redefining their reference point from their baseline visit, not from their final visit.

Using the previously described time imputation process, all subjects ended with an unambiguous reference point and were all properly aligned in time; this adjusted time variable could then be used to longitudinally characterize different features, including the 200 markers obtained from the image processing analysis. However, time imputation can impact the final shape of the mean value curves over time, due to the fact that the number of subjects that can now be time-aligned increases significantly, from 3909 to 6570. As such, what followed was a procedure to ensure that the temporal characterization of these variables was not significantly dependent on the conditions set by the imputation process. For this, an almost identical process to the previously defined time imputations was executed, with the exception that instead of using the highest ranked instance to create the new reference point, it was randomly selected from one of the top five instances. This process was repeated 1000 times, thus generating 1000 different time variables that could be used to characterize the markers. As such, the data for one marker can be longitudinally arranged in 1000 different ways using the available time data.

The original imputed time variable was compared with each one of these 1000 validation time variables using a two-sample bootstrapped Kolmogorov–Smirnov (KS) test using the Matching library for R developed by Jasjeet (version 4.9.9), with a null hypothesis declaring that both samples follow the same distribution [12]. A significance level α of 0.05 was chosen.

### 2.5. Longitudinal Analysis

The longitudinal characterization of each feature of interest was performed using a spline regression model, with a maximum of one algorithmically calculated knot per model. The use of spline regressions follows the assumption that the features being evaluated should have a sigmoid behavior in time when describing them from normal cognition to dementia, which is shown in models such as the one developed by Jack et al. [2]. The sigmoid curve can be simplified into three straight lines; each extreme of the curve is represented by a separate line, with an intermediate line connecting them. Furthermore, since we are only modeling the time between MCI and AD, only a subsection of the sigmoid should be observed, and therefore curves were set to only have one knot; that is, they were to be made from two straight lines.

Spline regression models were generated with the help of the Segmented R package developed by Vito and Muggeo (version 1.3.4), which iteratively estimates the position in time of the knot and the slopes of the linear relationships observed in the data, using a negative log-likelihood as the objective function to be minimized [13]. For these models, it is expected that the position of the knot indicates the time in which the most significant change in tendencies occurs. Outliers were excluded from the data points used to generate the regression models; these were defined as the values that lay outside the range of 1.5 times the interquartile range (IQR) below the first quartile, and 1.5 times the IQR above the third quartile.

## 3. Results

Following the time imputation process, for each set of generated times (including the original), a curve of average values over time for a particular variable was generated. In order to illustrate the results of this procedure, graphs were generated of all the curves for the MMSE score and mV of the medial and inferior temporal gyrus as segmented by the MAPER algorithm; these are shown in Figure 2. 

Table 2 shows the mean, minimum, and standard deviation of the p-values given by the two-sided KS tests used to evaluate the generated curves to a series of 12 randomly selected features. These tests were not performed for all variables due to a lack of computational power.

What follows are figures showing the results of the spline regression models. Figure 3 shows typical cases of model results. On each of the four graphs, the measured variables show two different tendencies as time progresses; the point in time in which one tendency changes into the other is indicated by the knot of the regression model. The positions of the knots (in months) estimated by the spline regression algorithm are −96.00 (mSA, hippocampus), −47.99 (mV, middle and inferior temporal gyrus), −6.00 (dV, lateral orbital gyrus), and −48.00 (Jaccard index, anterior temporal lobe). It should be noted that the knot might indicate different phenomena in each of the variables; for some variables, such as the mean surface area of the hippocampus, the knot indicates the moment in which the variable begins to stabilize. For others, such as the mean volume of the middle and inferior gyrus, the knot indicates the moment at which the variable begins to destabilize.

Figure 4 shows examples of inconclusive model results due to a lack of changes in tendency on time shown in the available data. Unlike the results from Figure 3, the knots of these regression models cannot be confidently given a meaningful interpretation.

Figure 5 shows the regression results to AD markers obtained directly from the ADNI database. The knots’ estimated positions in time (measured in months) for the variables shown are as follows: −44.11 (MMSE Score), −62.65 (ADNI-MEM), −92.37 (Tau CSF concentration), and 24.00 (Aβ CSF concentration). For the MMSE score, the ADNI-MEM score, and the Tau concentration variables, the knot indicates the point in time in which the measurements begin to destabilize. On the other hand, the knot generated for the Aβ concentration variable indicates the point in time in which the measurement begins to stabilize. Following the model of biomarker evolution proposed by Jack et al., our results generated using the spline regression model showed that the knots shared similar temporal order, where the Aβ concentration is the first one to stabilize followed by the Tau concentration, volumetry, and finally clinical function represented by MMSE and ADNI-MEM [2].

Plots showing the results for all the spline regression models we developed can be found in the Appendix A to this article.

The results of the regression models were integrated into Figure 6, which attempts to reproduce the proposed model of biomarker evolution proposed by Jack et al., while adding the Jaccard Index of the gyri parahippocampalis [2].

## 4. Discussion

Adjusting the time reference of patients allows us to analyze the behavior of the different markers throughout the patient’s visits and therefore a longitudinal characterization of them. Although this was already possible before the imputation of times for variables curated directly from the ADNI database, the imputed times are required for variables acquired from MRI studies. This is because all the studies used were taken on the patient’s first visit; therefore, for these variables, there will only be information available associated with a diagnosis of AD if the patient who underwent the MRI study received a diagnosis of AD on their first visit. Thanks to the time imputation process, there exists information for subsequent visits. Computing these average value curves such as the ones shown in Figure 3, Figure 4 and Figure 5, a longitudinal characterization of the desired variables can be carried out, analyzing the presence of trends and points where significant changes occur.

The p-values shown in Table 2 suggest that there is no significant difference between the distribution of all the curves generated through imputation. Therefore, the original set of imputed times is suitable for use in the longitudinal characterization of the variables to be evaluated.

In the regression models created, some common trends can be noticed. In particular, the two most common cases are the following: the variable goes from showing a high rate of change to a stage of stability, or vice versa. The MMSE and ADNI-MEM scores, both shown in Figure 5 show a combination of both trends, approximating a sigmoid curve longitudinally. Since only one knot per variable was specified, the selected knot corresponded to the first significant change in tendencies in both cases. If the second tendency found is stabilization, the knot is usually found after the first AD diagnosis. Figure 5 also shows the generated regression models for the Total Tau and P-Tau concentrations; due to a lack of information on these variables, only one extreme of the expected sigmoid curve is represented in the data, which is also shown in the graphs for the variables acquired through the MRI study analysis (Figure 3).

The general tendencies observed for each type of generated variable are as follows. Variables measuring averages in volume and surface area (mV and mSA) tend to decrease as time progresses. Similarly, the Jaccard index generally tends to decrease with time. The variables showing the opposite tendency are the ones measuring differences in volume and surface area (dV and dSA), which tend to increase in value with time. This indicates that regions will typically decrease in size as AD progresses, and that equivalent regions in contralateral hemispheres will be less similar. As such, in these cases, the region in one hemisphere will be more affected by the atrophy than the region in the opposite hemisphere. Examples of typical tendencies are shown in Figure 3.

Exceptions to the previously described typical tendencies were observed for each type of variable. Notable but expected exceptions to the tendencies observed in mV and mSA occur in the regions covering the lateral ventricles and the caudate nucleus. In these cases, the volume and surface area of these regions tend to increase over time, which corresponds with existing findings in the literature investigating these regions in patients with AD. A study investigating ventricular enlargement in healthy patients and patients with MCI and AD found that patients with AD report a significantly larger lateral ventricular volume in comparison to healthy patients [14]. Another study reported an increased caudate nucleus volume in patients with AD compared to healthy patients; however, it should be noted that other articles have reported opposite results for patients with AD [15,16]. As such, while our results complement the findings of Persson et al., we would not consider them to be conclusive and follow-up analyses with more data would be recommended in order to solidify these claims [15].

For the Jaccard Index, there are exceptions to the declining tendency found in the majority of regions, as shown by the cingulate gyrus (anterior part), lateral ventricles (temporal horns), and the pre-subgenual frontal cortex. These show an increase in the Jaccard index from the beginning, maintaining this increase for most of the time that data are available. This could be possibly explained by stronger atrophy in the larger region.

Exceptions to the typical trend observed in the difference-based variables (dV and dSA) take place as decreasing tendencies. Typically, this decrease is consistent, as in the nucleus accumbens for the difference in surface area. Assuming that this result is reliable, it could be that the previously less stunted region begins to lose volume at a higher rate than the other region, causing the difference in volumes to decrease. The trend in the anterior temporal lobe, middle part is also notable, since both dV and dSA show an increase followed by a decrease after the knot.

Figure 4 shows regression models where visual inspection of the data shows no noteworthy changes in tendency for the measured variables. The points in the graphs that were estimated using the biggest samples show the stability of the variable across the available time range. Following the assumption that the ideal behavior over time of these variables will be sigmoid-shaped, the time range covered by the data is insufficient to show the points in which significant changes in the variables occurred. In these cases, the most accentuated changes instead occur at the extreme ends of the time variable, where tendencies are noisier due to a lack of available data. As a result, the spline regression algorithm then fits a model based on the behavior on these points; since these are not backed up by sufficient data, we consider these models to inaccurately reflect any tendencies and as such were discarded from further analysis.

Figure 6 shows an attempt at recreating the biomarker model described by Jack et al. while adding the results for the gyri parahippocampalis’ Jaccard index; this region was chosen due to evidence showing that parahippocampal atrophy is a good differentiator between AD and MCI/healthy cases [2,17,18]. The resulting curves show similar behavior to the ones shown by their biomarker model; for instance, the β-Amyloid sigmoid curve reaches its peak around the point of AD diagnosis, and the MMSE sigmoid curve is the last one to reach its peak, occurring after the AD diagnosis. It should be noted that clinical function, which is integrated into their model, could not be represented in our model with the data available to us. The novel feature of Figure 6 is its Jaccard index curve: it peaks earlier than the other biomarker curves, which would suggest an increased capability to achieve an early AD diagnosis.

The main limitation of this project is the number of MRI studies available for analysis. The lack of data is noted when graphing mean longitudinal curves for the markers derived from analyzing these studies, since they present trends with considerably more noise than the longitudinal mean curves of variables such as the MMSE score where considerably more data are available, particularly at the extremes of the time range. The noise removes certainty from claims that could be derived from the obtained regression models; as such, a continuation of this analysis using a bigger number of MRI studies is recommended in order to obtain more properly sustained results. Performing this asymmetry analysis for follow-up studies of the patients, as well as acquiring studies of a bigger number of patients, could provide more information on the longitudinal development of these variables. The lack of stratification based on MCI progression could also be a source of noise for the analyzed variables, since no distinction was made between patients whose cognitive impairment remains mild with stable, non-degenerating symptoms, and patients who eventually develop more severe forms of dementia. This and other forms of stratification, such as analyzing patients separately based on the presence of other symptoms or test results, could improve our results and can be achieved with an increase in the available MRI studies in order to have sufficient data available in each stratified set.

Another limitation is the lack of data available on the extreme ranges of the time variable, which leads to erroneous regression results such as the ones shown in Figure 5. As such, ignoring these data points in future analyses is advisable in order to potentially improve the results, although increasing the number of analyzed MRI studies would be recommended in order to compensate for the ignored data. Finally, another hindrance to the methodology is the lack of a standardized process to evaluate the final transformation of the registration in the asymmetry analysis; the size of the analyzed regions impacts the magnitude of the final metric achieved, and this makes the efficacy of the registration processes difficult to compare directly between each pair of regions.

## 5. Conclusions

By comparing the commonly used biomarkers with our proposed asymmetry-based markers, we can appreciate that our methodology for spline regression not only shows similar results to those currently available in the literature, but also demonstrates that some regions’ asymmetry shares the same behavior, suggesting their potential as a novel biomarker for Alzheimer’s diagnosis. Additionally, the results suggest that further investigations into measures of asymmetry are a worthwhile endeavor in order to improve efforts to achieve an early diagnosis of the disease.

## Figures and Tables

**Figure 1 healthcare-10-01643-f001:**
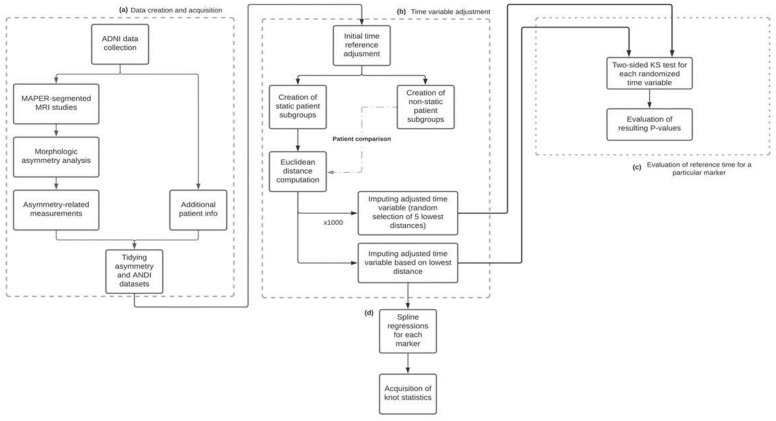
Flowchart illustrates the following methodology: (**a**) Data creation and acquisition: generating a tidy dataset. (**b**) Time adjustment: time reference adjustment for valid longitudinal characterization. (**c**) Evaluation of reference time: evaluation between the generated curve and the original using the KS test. (**d**) Spline regression: each marker with the resulting knots shows its most significant inflection point.

**Figure 2 healthcare-10-01643-f002:**
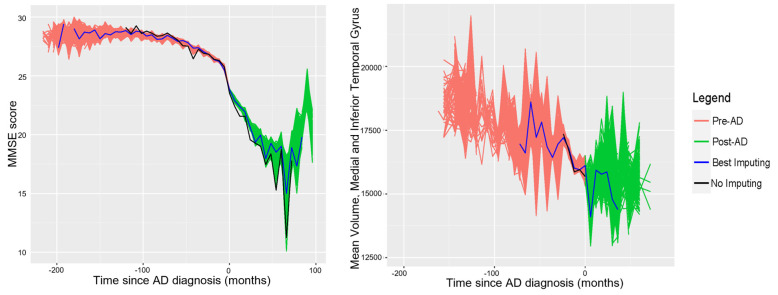
Average longitudinal curves for the MMSE Score and mean volume of the medial and lateral temporal gyrus using all generated time variables. In order to eliminate unwanted noise from these curves, the data points with less than 5 measurements available on a particular value of time were not plotted.

**Figure 3 healthcare-10-01643-f003:**
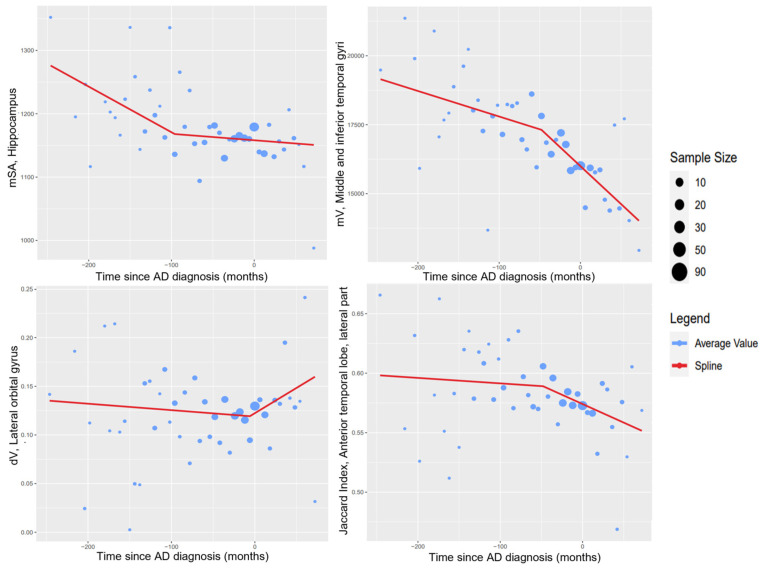
Examples of spline regression model results.

**Figure 4 healthcare-10-01643-f004:**
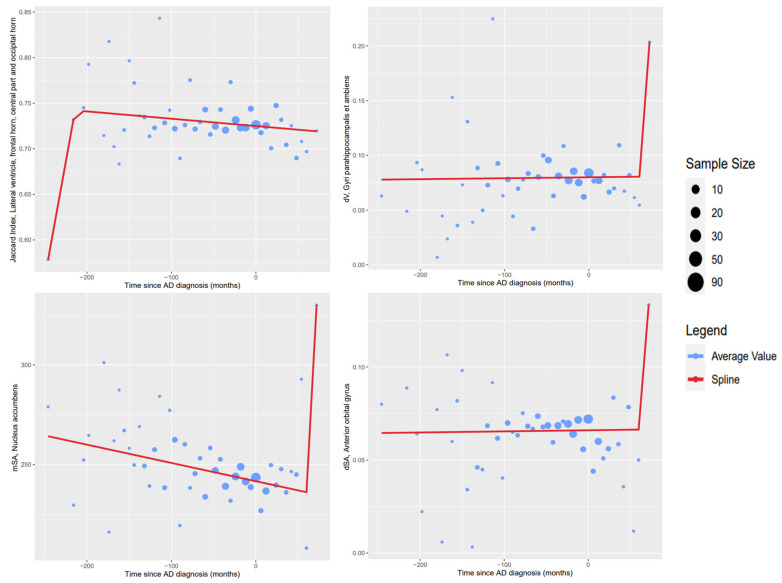
Examples of data without noteworthy changes in tendency in the measured variable.

**Figure 5 healthcare-10-01643-f005:**
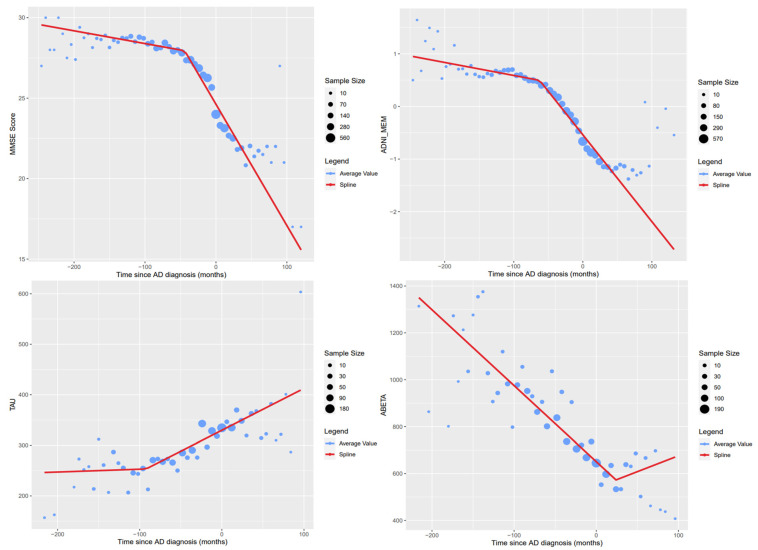
Spline regression results for commonly used AD markers.

**Figure 6 healthcare-10-01643-f006:**
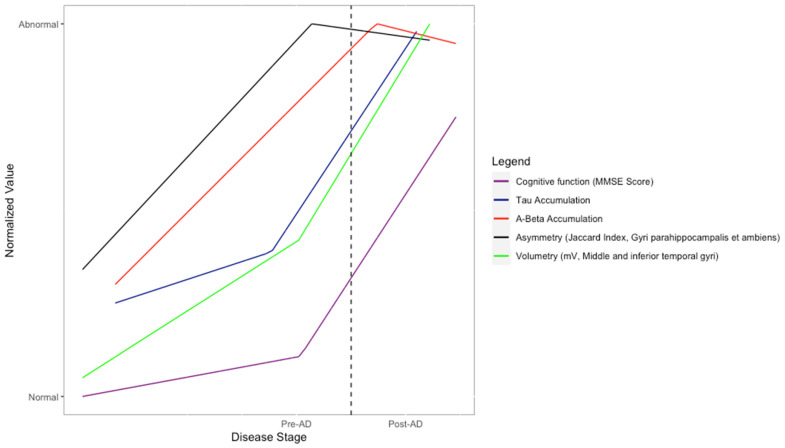
Comparison of different spline regression models, adjusted to show all of them indicating an increase over time for visualization purposes.

**Table 1 healthcare-10-01643-t001:** Summary of patient characteristics analyzed in the study.

Characteristic	Value (*n* = 6529)
Number of patients	1382
Number of men (and percentage)	808 (58.47%)
Number of women (and percentage)	574 (41.53%)
Average MMSE score (and range)	25.36 (0–30)
Average ADNI-MEM score (and range)	−0.20 (−2.90 to 3.20)
Average patient age at baseline (and range)	73.76 (54–91)
Average Tau concentration in pg/mL (and range)	320.36 (97.31–915.80)
Average P-Tau concentration in pg/mL (and range)	31.21 (8.21–103.70)
Average Aβ concentration in pg/mL (and range)	854.39 (210.90–3331.00)

**Table 2 healthcare-10-01643-t002:** P-Values calculated by the two-sided Kolmogorov–Smirnov tests.

Variable	Minimum	Mean	Std. Dev.
mSA (Lateral ventricle, frontal horn, central part, and occipital horn)	0.494	0.974	0.047
mV (Posterior orbital gyrus)	0.612	0.972	0.052
dV (Anterior temporal lobe, lateral part)	0.602	0.971	0.053
mSA (Subcallosal area)	0.62	0.970	0.052
Jaccard (Cuneus)	0.566	0.968	0.056
dSA (Anterior orbital gyrus)	0.427	0.966	0.062
mV (Postcentral gyrus)	0.451	0.958	0.070
Jaccard (Subcallosal area)	0.44	0.955	0.073
dSA (Superior parietal gyrus)	0.417	0.938	0.091
ADNI-MEM	0.343	0.935	0.110
mSA (Medial orbital gyrus)	0.419	0.910	0.114
mV (Inferior frontal gyrus left)	0.386	0.899	0.115

## Data Availability

Data used in the preparation of this article were obtained from the ADNI database (adni.loni.usc.edu).

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
