# Peer review of "Fitting Contralateral Neuroanatomical Asymmetry into the Amyloid Cascade Hypothesis"

_healthcare, 2022, doi:10.3390/healthcare10091643_

Round 1

Reviewer 1 Report

In this manuscript, the authors proposed that measuring contralateral neuroanatomical asymmetry could be a useful strategy for an early diagnosis of AD. They obtained the temporal evolution of the asymmetry measures and other general markers of AD by generating spline regression models using the data from ADNI database, and found asymmetry measures in particular regions showed significant changes earlier than any other general markers of AD.

I think their findings are interesting and potentially useful for the development of the AD diagnosis, although further validation is required for the approach. I therefore estimate the manuscript substantially contains results suitable for publication.

I listed below my individual comments.

  It would be better to include the conclusion of this study in the title.

  It is difficult to see almost all the labels in figures and lines in graphs. Use larger fonts and adjust the line thicknesses and colors.

  Figure 3 and 4 only show several examples of model results. It would be useful if readers can access all of their results in supplemental figures or somewhere else.

  Figure 6 only shows the result of Gyri parahippcampalis for the Jaccard Index. Is it possible to provide the results of all the other regions in supplemental figures or somewhere else?

  Line34, func-tional -> functional

  Line37, diagno-sis -> diagnosis

  Line274-275, delete ‘begins to stabilize’ at the end.

Author Response

First of all thank you for your observations, we used them to improve our work. 

The figures 3, 4, and 5 were modified to improve legibility, with a bigger font size and line thickness. A supplementary file with all the spline regression model results was also created, to make these results accessible to interested readers; this one also includes all our results for the Jaccard Index. The orthography issues you pointed out were also corrected.

Reviewer 2 Report

Comments to the Authors

Authors have evaluated novel diagnostic methods through determination of contralateral neuroanatomical symmetry. The study seems quite interesting and could be beneficial in future perspective to find out therapeutic approaches for AD which is lacking any concreate treatment. I just have few points to comment on.

1)      Symmetry-related measures is the biomarker or surrogate biomarker?

2)      Can the authors mention the novelty and hypothesis more elaboratively in the last part of the Introduction section?

3)      Were the patients of uniform age, height, and weight? Because these parameters might influence the brain size.

4)      Discussion can be corroborated with previously reported studies.

5)      Is there any possibility of presence other types of dementia in addition to AD with the current diagnosis method?

Minor

1)      In line 37 correct the word diagnosis and amyloid in line 38, dysfunction in line 52, and etc.

2)      Can the texts in Fig. 1 can be more legible?

Author Response

Good day, thank you for your observations; we took them into account to further improve our manuscript. To comment on your observations:

1)      Symmetry-related measures is the biomarker or surrogate biomarker?

From our perspective, these biomarkers are not surrogates; this is why we don't name them as such in our manuscript.

2)      Can the authors mention the novelty and hypothesis more elaboratively in the last part of the Introduction section?

We redacted the last paragraph of our Introduction to further emphasize our hypothesis and the novelty of our approach compared to more traditional biomarkers used for AD diagnosis.

3)      Were the patients of uniform age, height, and weight? Because these parameters might influence the brain size.

The patients were not of uniform age, height, and weight. In order to handle the potential influence of these parameters in the brain size, we used the methodology shown in lines 178-180. Furthermore, in our discussion, we do propose that follow-up studies with data stratified on measurements such as these would be valuable, in order to better control the impact these measurements can have over the final results.

4)      Discussion can be corroborated with previously reported studies.

In our discussion section we currently corroborate our results with the following studies:

(Echávarri et al., 2010) (Dalboni et al., 2020) (Persson et al., 2018) (Jiji et al., 2013) (Apostolova et al., 2012)

5)      Is there any possibility of presence other types of dementia in addition to AD with the current diagnosis method?

The ADNI database from where the diagnostic information was extracted has strict protocols and inclusion criteria to avoid mixing other types of dementia in the studies.  As such, the possibility of presence other types of dementia in addition to AD in our data is unlikely.